# Mitigating Biases in Hate Speech Detection from A Causal Perspective

**Zhehao Zhang**[⋆]    **Jiaao Chen**[†]    **Diyi Yang**[◇]

[⋆] Dartmouth College    [†] Georgia Institute of Technology    [◇] Stanford University

zhehao.zhang.gr@dartmouth.edu    jiaaochen@gatech.edu

diyiy@cs.stanford.edu

## Abstract

*Warning: This paper discusses and contains offensive or upsetting content.*

Nowadays, many hate speech detectors are built to automatically detect hateful content. However, their training sets are sometimes skewed towards certain stereotypes (e.g., race or religion-related). As a result, the detectors are prone to depend on some short-cuts for predictions. Previous works mainly focus on token-level analysis and heavily rely on human experts' annotations to identify spurious correlations, which is not only costly but also incapable of discovering higher-level artifacts. In this work, we use grammar induction to find grammar patterns for hate speech and analyze this phenomenon from a causal perspective. Concretely, we categorize and verify different biases based on their spuriousness and influence on the model prediction. Then, we propose two mitigation approaches including Multi-Task Intervention and Data-Specific Intervention based on these confounders. Experiments conducted on 9 hate speech datasets demonstrate the effectiveness of our approaches. The code is available at https://github.com/SALT-NLP/Bias_Hate_Causal.

## 1 Introduction

Over the past few years, hate speech on social media has grown significantly in different forms. This causes serious consequences and societal impact on victims of all demographics (Mathew et al., 2021), largely affects their mental health (Saha et al., 2019) and even triggers real-world hate crimes (Relia et al., 2019). As a result, automated detection of hate speech becomes especially important and there have been numerous research studies conducted to characterize and detect these hateful or toxic contents (Zhou et al., 2021; Lahnala et al., 2022; Kotarcic et al., 2022).

However, current hate speech detection approaches often fail to be robust and generalizable,

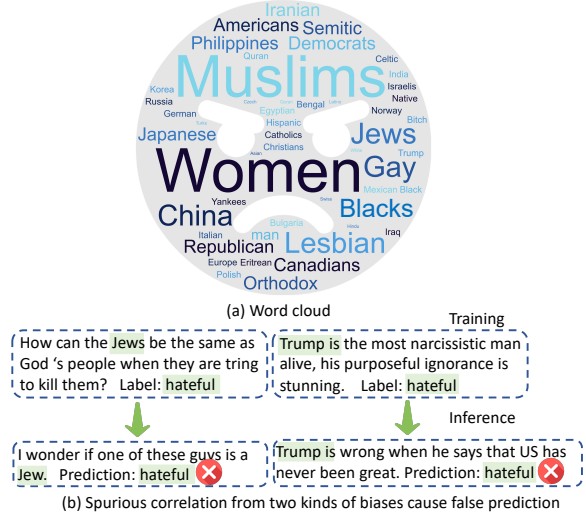

Figure 1: (a) Word cloud of frequent words in 9 widely used hate speech datasets and an example of how spurious correlations lead the model to make mistakes. The size of a word denotes its frequency of occurrence. (b) An example from Mandl et al., 2019 indicates that both token-level and sentence-level biases can introduce spurious correlations.

even to slightly different datasets on the same task, which largely prevents them from being applied in real-world applications (Vidgen et al., 2019). This might be due to the fact that current models are suffering from the spurious correlations between training data and labels (*e.g.,* hateful) (Ramponi and Tonelli, 2022), which might lead to the biased treatment of vulnerable and minority groups such as African American Vernacular English Speakers and may exacerbate racism (Harris et al., 2022a). One example is shown in Figure 1. The frequent co-occurrence of identity words and the hateful label in the training set can bias the detectors (e.g., fine-tuned BERT) to make false predictions during inference. As a result, it is of great need to comprehensively understand and identify the spurious correlations in hate speech detection and further mitigate the bias caused by spurious correlations.

A growing amount of recent work has examined the spurious correlation in hate speech detection

(Zhou et al., 2021; Kennedy et al., 2020; Sap et al., 2022), and found that a large number of tokens that target minority groups are highly correlated with the hateful label (Bender et al., 2021). As a result, methods like masking and removal of such tokens with the help of human annotations (Ramponi and Tonelli, 2022) have been proposed to mitigate these spurious correlations, together with finetuned large language models to generate artifacts to augment the training (Wullach et al., 2021; Hartvigsen et al., 2022). Despite these successes, they usually focus on identifying *token-level* spurious patterns in *one specific dataset manually* (Bender et al., 2021) while neglecting spurious correlations beyond tokens (i.e., in sentence levels, shown in the right part of Figure 1(b)) and lacking comprehensive studies across different datasets and domains. Moreover, these mitigations are often costly and time-consuming as human annotation and fine-tuning large language models to generate numerous data are required. Thus, a systematic study is needed to first automatically identify spurious correlations in a given hate speech dataset and then effectively and efficiently mitigate them.

To fill in this gap, we first conduct a comprehensive analysis to discover spurious correlations from both token-level and sentence-level on 9 hate speech datasets in an automatic way. Specifically, we use mutual information with domain knowledge to identify token-level spurious correlations such as identity words and leverage context-free grammar to investigate sentence-level highly correlated grammar patterns. We further propose a novel metric called Relative Spuriousness (RS) to verify the spuriousness of discovered spurious correlations based on its influences on model prediction. Our analysis shows that token-level spurious patterns are usually more general that exist in almost all datasets while sentence-level spurious patterns are more dataset-specific. We further study how spurious correlations cause model bias from a casual perspective (as shown in Figure 2). one affecting the distribution between PLM's pretraining data and vulnerable identities, and another influencing the distribution between vulnerable identities and their context in hate speech datasets. To mitigate these two biases, we propose Multi-Task Intervention (MTI) and Data-Specific Intervention (DSI) for bias mitigation. MTI tries to mitigate biases from the pre-training corpus through training auxiliary tasks, while DSI focuses on eliminating biases

originating from limited data and contains a framework that automatically detects, validates, and mitigates biases through a counterfactual generator. Experiments conducted on 9 hate speech datasets and out-of-domain challenge sets demonstrate the effectiveness and robustness of our proposal.

To summarize, our contributions are: (1) We automatically identify spurious correlations and comprehensively analyze them in hate speech detection from both *token-level* and *sentence-level* across 9 datasets. (2) We introduce a novel metric, *Relative Spuriousness*, to evaluate the spuriousness of identified spurious correlations at the sentence level. (3) We study the bias caused by spurious correlations from a causal perspective. (4) We propose two strategies, MTI and DSI, to mitigate the biases and show consistent improvements in 9 datasets.

## 2   Related Work

**Debias Hate Speech Detection**   Recent works (Yin and Zubiaga 2021; Wiegand et al. 2019; Kennedy et al., 2020; Ma et al., 2020; Gehman et al., 2020; Sap et al., 2020; Dreier et al., 2022; Stanovsky et al., 2019; Sridhar and Yang, 2022; Thakur et al., 2023; Ziems et al., 2023) have been studying the generalizability and biases for hate speech detection (Talat et al., 2018; AlKhamissi et al., 2022; Röttger et al., 2022; Bianchi et al., 2022). For instance, prior work found that existing hate speech detection models are biased against African American Vernacular English Speakers (Harris et al., 2022b; Sap et al., 2019) and certain identity words are highly correlated with these hateful labels (Bender et al., 2021; ElSherief et al., 2021a). A group of data augmentation methods (Sen et al. 2021; Sen et al., 2022; Hartvigsen et al. 2022) are proposed to mitigate the biases. For instance, Wullach et al. propose a simple data augmentation method using a finetuned GPT-2 model to generate 100k hate and non-hate data. Ramponi and Tonelli find highly correlated tokens and manually categorize them into different groups, and further mask or remove these annotated spurious artifacts. However, human annotations are not practical in large-scale or multi-platform scenarios. Different from these prior works that mainly focus on data-specific biases and mitigation methods, our method automatically finds these shortcuts and proposes Multi-Task Interventions on model levels.

## 2.1 Spurious Correlation in NLP

Increasing attention has been focused on spurious correlations in NLP tasks (Tu et al., 2020; McCoy et al., 2019; Jia and Liang, 2017; Niu et al., 2020; Wang and Culotta, 2020; Vaidya et al., 2020; Clark et al., 2020; Du et al., 2021). A line of work tries to evaluate the models' robustness on pre-defined shortcuts by proposing challenge test datasets (McCoy et al., 2019; Rosenman et al., 2020; Röttger et al., 2021) or finding salient words (Simonyan et al., 2013; Pezeshkpour et al., 2022; Shrikumar et al., 2017; Han et al., 2020). Another line of work also intends to verify the spuriousness of extracted features. Gardner et al., 2021 suggests that all correlations between labels and low-level features are spurious. Eisenstein, 2022 claims these correlations will naturally appear in the majority of classification tasks and that domain knowledge is required to identify any correlations that may be harmful. Joshi et al. define the spuriousness based on the sufficiency of the feature and counterfactual intervention. However, their definition is based on an unbiased classifier to model the probability of sufficiency, which could not be practical in real applications. Besides, they do not use this metric to find spurious artifacts. Instead, they use domain knowledge to pre-define the potential feature. Furthermore, previous analyses (Zhang et al., 2018) of biases' influences on model predictions are limited. To this end, we propose a novel metric named Relative Spuriousness (RS) which considers the biases' influence on the model's decision-making.

## 3 Methods

This section describes how we identify, understand, and mitigate the spurious correlations in **H**ate **S**peech Detection (HS).

## 3.1 Identifying Spurious Correlations

Current HS models often suffer from spurious correlations (Ramponi and Tonelli, 2022; Bose et al., 2022; Wang et al., 2022): they tend to utilize prediction patterns that hold for the majority examples but do not hold in general. This might cause the model to be biased in applications (Ramponi and Tonelli, 2022; Gardner et al., 2021). In this section, we identify potential spurious correlations from two levels across multiple datasets.

### 3.1.1 Token-level Spurious Correlations

One specific label (e.g., *hateful*) might be highly correlated with certain tokens (e.g., "women") in the datasets (Wang et al., 2022; Ramponi and Tonelli, 2022). As a result, models might learn to make biased predictions only based on those token-level correlations while neglecting the whole semantic meaning. In order to identify such token-level spurious correlations, we utilize token-level PMI-based searching methods (Ramponi and Tonelli, 2022) to find biased words in HS. Specifically, following Ramponi and Tonelli, 2022, we compute the PMI using the equation described in Appendix A.1 between every word and the hateful label. After that, we select the highest correlated words for further investigation across 9 datasets. This led to a large set of *identity-related* words (over 80, shown in Figure 1 as an example) that are highly correlated with the hateful labels while the words themselves are neutral. We also observe that such identity words are often common across different datasets. Thus, we treat these identity words as *general* token-level spurious correlations[1].

### 3.1.2 Sentence-level Spurious Correlations

Beyond token-level identity words, certain sentence-level patterns might also be highly correlated with specific labels that might be spurious correlations. For example, in a platform with discussion on politics, patterns such as *The wall should* and *The wall is* are usually connected with discontent with the US border wall. A large number of such speeches are hateful. However, these grammar patterns are completely neutral. To automatically discover these sentence-level spurious correlations, following Friedman et al., 2022, we induce a grammar for HS training data and obtain the maximum likelihood trees in an unsupervised manner. Specifically, we use a probabilistic context-free grammar (PCFG), which contains the distinguished start symbol $\mathcal{S}$, terminal symbols $\mathcal{V}$ (words), non-terminal symbols $\mathcal{N}$ and the rules of the form $A \rightarrow B \in \mathcal{R}$, where $A \in \mathcal{N} \cup \mathcal{S}$ and $B \in \mathcal{N} \cup \mathcal{V}$. We use the same parameterization and training methods as Kim et al., 2019. We then compute the mutual information between grammar patterns (non-terminal root) and labels and find

---

[1] Note that this identification process is automatic and we manually inspect it to ensure its quality. To analyze such spurious correlations more comprehensively and reduce noises, we adopt another module (a pre-trained NER model) to ensure these highly-correlated words are identity-related.

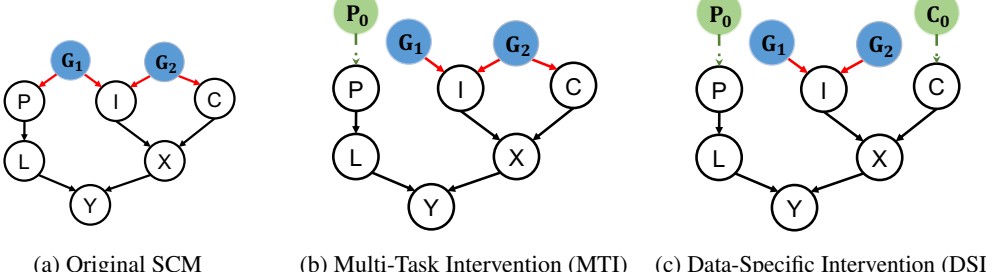

|                          |                          |                          |
|--------------------------|--------------------------|--------------------------|
| (a) Original SCM         | (b) Multi-Task Intervention (MTI) | (c) Data-Specific Intervention (DSI) |

Figure 2: SCM (Structural Causal Mode) for HS detection with vulnerable identities. We have omitted the unmeasured variable $U_*$ for each variable for brevity. $P \rightarrow L$: Using corpus $P$ to pretrain the language model $L$. $(I, C) \rightarrow X$: Vulnerable objectives $I$ (e.g., identity words) and their contexts $C$ constitute the input data $X$. $(L, X) \rightarrow Y$: Input the language model $L$ with data $X$ to output the prediction $Y$. (a) Two confounders $G_1$ and $G_2$ bias the generation distribution of pretraining corpus and HS data. (b) MTI intervenes with the bias from $G_1$, where $P_0$ represents the training corpus of auxiliary tasks. (c) DSI further intervenes with the bias from $G_2$, where $C_0$ denotes the newly generated context. To analyze grammar patterns biases, we only need to change $I$ and other parts of the SCM remain the same.

highly-correlated ones. We define the patterns in terms of grammar subtrees. After we apply the above method to 9 hate speech datasets, we find that different datasets have different highly correlated patterns (Section 4.5.1 as examples). As a result, such patterns can be considered potential *data-specific* spurious correlations.

**Spuriousness Validation** After discovering these sentence-level spurious correlations candidates, we further validate their spuriousness to identify clean spurious correlations. We define Relative Spuriousness (RS) based on the necessity of a feature's influence on the model's prediction. Our definition depends on two properties of spurious features: a feature's existence significantly impacts a specific label on a biased model, and it is not important for an unbiased model ideally.

**Definition 1** (Relative Spuriousness (RS)) The RS of a feature $x_i$ for the label $y$ is:

$$P_{RS} = D_l(x_i, y) - D_g(x_i, y),$$
$$D_l(x_i, y) = P_l(Y = y | X_i = x_i, Y = y)$$
$$\quad - P_l(Y(X \neq x_i) = y | X_i = x_i, Y = y),$$
$$D_g(x_i, y) = P_g(Y = y | X_i = x_i, Y = y)$$
$$\quad - P_g(Y(X \neq x_i) = y | X_i = x_i, Y = y).$$

Here $P_l$ indicates the local probability, where we use the HS detector's softmax output trained on a single HS dataset to model it. $P_g$ represents the global one, where we use the average response of models trained on every HS dataset. Note that we do not use the model trained on all datasets for the class-imbalance problems. $D_l$ and $D_g$ mean the local and global probability difference between HS examples with and without feature $x_i$, respectively. Intuitively, for spurious features, it can have

a great impact on local probability and almost no impact on global probability. Therefore, it has large $D_l(x_i, y)$ and small $D_g(x_i, y)$, causing higher RS. Compared with the prior work (Joshi et al., 2022), we consider a feature's relative influence on the model's prediction locally and globally, filling in the gap of considering features other than $x_i$, which is a reasonable and practical metric to find spurious features. We select the features that have RS higher than a certain threshold as data-specific biases. The following sections investigate the origin of the above biases and propose two intervention approaches to mitigate them.

### 3.2 Understanding Spurious Correlations

With the identified spurious correlations in HS, it is then of great need to study how they might be generated and how the models might suffer from them by making biased predictions. In this section, we visualize the relations between spurious correlations and the caused bias from a causal perspective (Feder et al., 2021; Hardt et al., 2016; Kilbertus et al., 2017; Vig et al., 2020). Specifically, we use Structural Causal Model (SCM) (Pearl et al., 2000) which is a conceptual model that describes the causal mechanisms of a system, to recognize the confounders which are variables that influence both the dependent variable and independent variable, causing spurious correlations. We use directed acyclic graphs (DAGs) $G = \{V, f\}$ to describe the causal relationships between different variables. $V$ refers to variable nodes and edges denote causal relation function $f$.

We visualize the SCM for HS detection described in Figure 2a, where identities $I$ and the corresponding context $C$ constitute the input data $X$. The

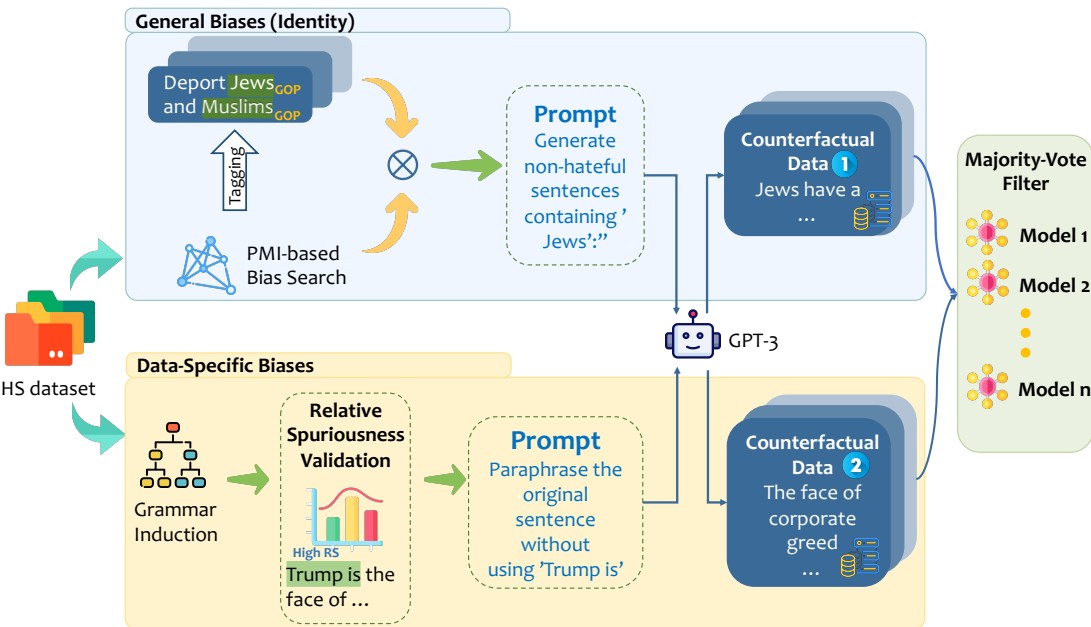

Figure 3: Our proposed pipeline (DSI) is to find spurious features and mitigate these biases. The blue part indicates the framework for token-level biases (identity). The yellow part represents the framework data-specific biases. ⊗ indicates intersection. For all the biases, we use off-the-shell GPT-3 as the counterfactual generator followed by a majority-voting filter. We use the counterfactual data along with the original training data to train the HS detector.

evaluation result $Y$ (accuracy or $F_1$ score) is generated by PLM $L$ and input $X$.

**Confounders** Two confounders $G_1$ and $G_2$ influence their correlated variable's independence of generation distribution, causing spurious correlations. The two confounders are as follows.

- $G_1$: is the confounder which affects the distribution of both $P$ and vulnerable identities $I$. For example, the unbalance appearance frequencies of different races of people in the pretraining data for the pretrained language models may constitute this confounder.

- $G_2$: influences the distribution of vulnerable objectives $I$ and their context $C$. The training data's source may compose this confounder. For instance, de Gibert et al. 2018 collect data from Stormfront, a white supremacist forum, where mentions of people of color (POC) usually occur in hateful contexts.

### 3.3 Mitigating the Bias in HS

Based on the above two confounders that cause spurious correlations, we then propose two intervention techniques to mitigate the bias in HS.

**Multi-Task Intervention (MTI)** The motivation for MTI is twofold. Firstly, MTI tries to mitigate the spurious correlations between PLM's training

corpus $P$ and vulnerable objectives $I$ (caused by $G_1$) in order to resolve two kinds of biases in Section 3.1. By training with MTI objectives on a wide range of HS, we expect that the original distribution connection of $P$ and $I$ can be altered. Secondly, most HS is composed of unregulated sentences. MTI can make the PLM get a more robust representation for HS. Specifically, we introduce two auxiliary tasks on PLM to alleviate the bias from the pre-training corpus: (1) **Masked Language Modeling (MLM)**(Devlin et al., 2019): we randomly mask 15 % tokens from hateful sentences from 9 HS datasets and let PLM predict the tokens on the masking position with language modeling objective. (2) **Multi-Task Learning (MTL)**: We define different tasks as different hate speech datasets $D_i = \{x_i^j, y_i^j\}_{j=1}^{n_i}$ and train on all the datasets $\{D_i\}_{i=1}^{N}$ (see Section 4.1 for details on datasets) simultaneously using the same PLM and different classification heads.

**Data-Specific Intervention (DSI)** To mitigate the influence of $G_2$ and inspired by current progress in counterfactual learning (Zeng et al., 2020; Sen et al., 2021; Eisenstein, 2022; Garg et al., 2019; Davani et al., 2021), we propose a counterfactual generator for DSI. Recent works (Mishra et al., 2022; Ouyang et al., 2022) discover that the powerful Large Language Models (LLMs) (Brown et al.,

| Method | MM | WSF | Fox | HASOC | ETHOS | TSA | AHS | AOH | ZH | Average |
|---|---|---|---|---|---|---|---|---|---|---|
| `davinci-002` zero | 48.64 | 87.13 | 55.81 | 66.67 | 34.15 | 75.00 | 44.44 | 80.00 | **80.81** | 63.63 |
| `davinci-002` few | 41.86 | 86.87 | 45.61 | 66.67 | 77.19 | 31.58 | 60.00 | 75.00 | 70.97 | 61.75 |
| `davinci-003` zero | 48.86 | 59.57 | 74.43 | 68.25 | 84.48 | 92.42 | 63.06 | 78.51 | 40.26 | 67.76 |
| `davinci-003` few | 53.13 | 73.33 | 61.22 | 63.28 | **86.29** | 86.61 | 73.33 | 79.48 | 35.21 | 67.98 |
| ChatGPT zero | 52.26 | 63.94 | 46.54 | 52.38 | 54.73 | 69.81 | 33.46 | 43.33 | 17.89 | 48.26 |
| ChatGPT few | 33.95 | 40.46 | 35.44 | 53.13 | 55.50 | 89.57 | 40.95 | 46.86 | 34.78 | 47.84 |
| Finetune | 48.77 | 77.49 | 71.22 | 65.27 | 77.77 | 88.48 | 93.33 | 88.42 | 67.33 | 75.34 |
| + MLM † | 43.12 | 77.49 | 70.88 | 64.49 | 76.18 | 87.94 | 93.21 | 89.67 | 68.67 | 74.63 |
| + MTL † | **61.63** | **89.65** | **84.77** | **80.28** | 76.44 | **93.58** | **96.06** | **93.89** | 79.21 | **83.95** |

Table 1: Experiment results (macro-$F_1$ score) of two GPT baselines and BERT finetune with two MLI methods on 9 HS datasets. All Finetune results are averaged over three runs using three random seeds. GPT-3's performances vary greatly across datasets. MTL has an 8.61 $F_1$ improvement on average compared to Finetune. † means our method.

| Method | MM | WSF | Fox | HASOC | ETHOS | TSA | AHS | AOH | ZH | Average |
|---|---|---|---|---|---|---|---|---|---|---|
| Back-Trans$^\triangle$ | 60.65 | 90.85 | 83.41 | 79.73 | 76.86 | 92.67 | 94.39 | 93.40 | 78.82 | 83.42 |
| Span-cut$^\triangledown$ | 58.93 | 91.10 | 84.25 | **81.71** | 88.85 | 92.86 | 95.58 | 92.30 | 79.53 | 85.01 |
| Token-cut$^\triangledown$ | 59.10 | **91.83** | 83.86 | 81.30 | 88.73 | 92.57 | 95.69 | 92.46 | 79.24 | 84.98 |
| Feature-cut$^\triangledown$ | 58.54 | 91.37 | 86.36 | 81.13 | 88.89 | 92.66 | 96.44 | 92.83 | 80.33 | 85.39 |
| AEDA$^\blacktriangle$ | **62.71** | 90.12 | 82.14 | 77.50 | 74.02 | 92.54 | **97.23** | 93.79 | 76.76 | 82.98 |
| Masking$^\blacktriangle$ | 58.31 | 90.65 | 83.05 | 80.47 | 87.71 | 92.06 | 95.11 | 93.37 | 80.49 | 84.58 |
| Removal$^\blacktriangle$ | 59.78 | 90.58 | 84.68 | 81.13 | 89.71 | 92.95 | 96.18 | 91.98 | **80.94** | 85.33 |
| DSI † | **62.71** | 91.65 | **87.04** | 81.39 | **90.76** | 93.17 | 96.52 | **93.82** | 80.65 | **86.41** |

Table 2: Experiment results (macro-$F_1$ score) of token-level($\blacktriangle$), sentence-level ($\triangle$), and hidden-level ($\triangledown$) data augmentation methods along with two token-level debias methods and DSI on 9 HS datasets. All these methods are based on MTI. All results are averaged over three runs using three random seeds. We conduct the significant test following Berg-Kirkpatrick et al., 2012 and the average estimate of p-value is 0.003 ($< 0.01$) between DSI and SOTA baselines, demonstrating the significant differences.

2020; Touvron et al., 2023; OpenAI, 2023b) with instruction prompt can have outstanding performances in many NLP tasks. Therefore, we use off-the-shell GPT-3 with instructions as the backbone of the counterfactual generator. The overall framework of DSI is illustrated in Figure 3. After we find spurious features using the method described in Section 3.1.2, we employ different instructional prompts for general biases (identity) and data-specific biases. The details on prompt design can be found in Appendix A.4. In practice, we find that the counterfactual data may not always remain the same label as described in the prompt. Thus we further conduct a majority voting with the models trained on different HS datasets. We use a majority vote to make sure the generated counterfactual data are non-hateful, which is a way to further control the quality of the generated data. We then train the HS detector with the original training data plus the counterfactual ones.

## 4 Experiment

### 4.1 Datasets and Setup

We conduct experiments on 9 datasets [2] including Multimodal Meme Dataset (text) (Suryawanshi et al.), Hate speech dataset from a white supremacist forum (de Gibert et al., 2018), Fox-News-User-Comments (Gao and Huang, 2017), HASOC19 (Mandl et al., 2019), ETHOS (Mollas et al., 2022), Twitter Sentiment Analysis (TSA), Anatomy of Online Hate (AOH) (Salminen et al., 2018), Hate Speech on Twitter (Zhang et al., 2018) and Hate-Offensive (AHS) (Davidson et al., 2017). Following previous work (Ramponi and Tonelli, 2022; ElSherief et al., 2021b), we adopt BERT-based-cased as the detector backbone model. To better avoid the issue of class imbalances, we use macro $F_1$ as the primary metric (other metrics such as RS are also explored in Section 4.5.2). More implementation details can be found in Appendix A.5.

---
[2] Statistics along with generated counterfactual data can be found in Appendix A.2.

## 4.2 Baselines

To evaluate the effectiveness of MTI, we compare them with the following baselines[3]: **Finetune** (Devlin et al., 2019): finetunes the BERT for hate speech detection (Tran et al., 2020). **text-davinci-002/text-davinci-003/ChatGPT zero/few** (Brown et al., 2020; OpenAI, 2023a): uses text-davinci-002/text-davinci-003/ChatGPT along with a text prompt to model the classification problem into a conditional generation problem (with a few examples). **MLM + Finetune**: first conducts Masked Language Modeling (MLM) (Devlin et al., 2019) on hateful speeches to get a more robust representation before finetuning. **MTL + Finetune** first conducts Multi-Task-Learning (MTL) before finetuning. Specifically, we train 9 datasets simultaneously with separate label space in MTL.

To evaluate the effectiveness of DSI, we use the following methods as baselines based on MTL + Finetune (best models): **AEDA** (Karimi et al., 2021): randomly inserts punctuation marks into the original text for data augmentation. **Back translation** (Edunov et al., 2018): first translates sentences into certain intermediate languages and then translates them back. **Cutoff** (Shen et al., 2020): masks tokens, spans, and features in the input space. **Artifacts-removal/ masking** (Ramponi and Tonelli, 2022): removes (or masks) any occurrence of spurious lexical artifacts (based on PMI) for training and validation data.

## 4.3 Main results

The main experiment results are shown in Table 1 and 2, respectively. As shown in Table 1, we observe that LLMs' performances on different datasets vary greatly. Besides, the performance of few-shot LLMs can not outperform zero-shot consistently, which is different from other tasks. As a result, these results indicate that *the off-the-shell LLMs is not always a good HS detector*, which is consistent with the results in (Ziems et al., 2023). One of the reasons behind this may be that HS's grammatical structure may not conform to the standard grammatical norms. For example, the authors of HS online hardly use complete sentences. As a result, HS datasets contain a large proportion of such unregulated sentences, which may confuse the off-the-shell GPT model. Among the LLMs we use, text-davinci-003 performs the best while ChatGPT has the worst performance. On the other hand,

---

[3]Implementation details can be found in Appendix A.6.2

---

| Methods | $F_1$ **score** |
|---|---|
| MTL | 82.53 |
| +AEDA | 81.87 |
| +Back-Trans | 81.19 |
| +Span-cut | 83.89 |
| +Token-cut | 82.54 |
| +Feature-cut | 82.55 |
| +Masking | 81.87 |
| +Removal | 79.01 |
| +DSI (Ours) | **88.49** |

Table 3: Experiment results of different methods on the challenge set. Our proposal DSI can have a more noticeable improvement over other baselines on OOD analysis.

the finetune-based model can learn these abnormal structures through training data. Furthermore, for the two MTI methods, MTL can further consistently improve the performance (8.61 $F_1$ score), indicating the importance of getting a robust representation of HS. Besides, MTL is orthogonal to other data-specific approaches (DA, data-specific debias methods). Therefore, the following experiments in Table 2 are all based on MTL.

From Table 2, we observe that our proposed DSI can outperform other baselines on average and have a noticeable improvement (2.46 $F_1$ score) over MTL. Besides, we find that token-level DA methods can even impair the overall performance, indicating that these methods are not as effective as in other tasks. On the other hand, three hidden-level DA methods do have a positive impact on the performance. Besides, previous debias methods such as artifact removal or masking also boost the performance but the improvement is subtle.

## 4.4 Generalization Analysis

Joshi et al. claim that building a "challenge set" to see if the intervention of the input cause model predictions to vary expectedly is a standard method of testing a model's robustness. Hence, to verify the robustness of our method, we construct an OOD challenge set using non-hateful counterfactual data generated by GPT-3 and hateful data from CONAN (Bonaldi, Helena and Dellantonio, Sara and Tekiroğlu, Serra Sinem and Guerini, Marco, 2022). Details on OOD datasets can be found in Appendix A.3. Given MTL's strong performances in Table 1, we evaluate other baselines based on MTL, as shown in Table 3. Our proposed DSI outperforms other methods by a larger margin than that in Ta-

ble 2, indicating the effectiveness of our method's robustness and great generalization ability.

### 4.5 Deep Dive of Sentence-level Biases

This part takes a deep dive into these sentence-level biases via qualitative analyses and visualization of their relative spuriousness distributions.

#### 4.5.1 Potential Data-Specific Biases

After conducting grammar induction and investigating 180 grammar patterns, we observe the following categories with high PMI with the hateful label: (1) **Absolute expression (8.8%)**: A large number of HS sentences contain absolute statements, where patterns like *all the*, *all other*, *any other,* etc. frequently occur. (2) **Hashtag before 'bad' words: (7.2%)** In online datasets, hashtags are common patterns. We observe that hashtags before some bad words (*e.g., # dickhead, # Liar, # Thief* etc.) are highly correlated with hateful labels. (3) **Aggressive actions (6.7%)**: A large number of HS contain radical action words (*e.g., fuck anything, kill yourself,* etc.). (4) **All capitalization (2.8%)**: All-caps patterns such as *FUCK YOU* and *HAVE NEVER* are more likely to be found in hateful sentences. In addition to the aforementioned types, there are other types of spurious patterns without cohesive themes in HS (see Table 6 for some examples). We also find that different datasets have different patterns. However, not all of the above patterns are spurious. The following section illustrates the RS distribution of these data-specific biases.

#### 4.5.2 Bias Distribution Analysis

For potential data-specific biases, we use RS described in Section 3.1.2 to validate their spuriousness. We visualize the distribution of these biases based on our proposed RS in Figure 4 in the Appendix. We find that most data-specific patterns' RS is positive, indicating that these highly-correlated artifacts do make the model more likely to predict a particular label. However, the experiment result shows that most of their RS is less than 0.2. As a result, most of the data-specific biases' impact on the HS detector's output is minimal. This is because most of these biased patterns do not contain hateful semantics. Although they are highly correlated with a specific label statistically, PLM can ignore them to some extent during inference.

### 4.6 Case Study of Token-Level Biases

To further analyze our proposal's effectiveness on identity biases, we analyze the errors made by the baseline model that is solved by our approach. As a result, we conduct the following case study on one of our 9 datasets used in this work, i.e., the white supremacist forum dataset (de Gibert et al., 2018). We randomly sample four false positive sentences where the baseline model makes mistakes in the test set. As shown in Table 4, the baseline model can not correctly identify some complex non-hateful examples containing identity words. For example, the sentences containing ethnicity-related words (e.g., *jews*, *Blacks*, *hebrews*, and *Arabs*) confuse the baseline detector. Besides, gender-related words (e.g., *women*) may also make the model more likely to classify the sentence as hateful. However, these sentences are non-hateful based on their neutral contexts. Inspired by saliency methods (Simonyan et al., 2013; Sundararajan et al., 2017; Smilkov et al., 2017; Balkir et al., 2022), we compute the probability differences between original sentences and identity-masked sentences to better interpret the model prediction. The average differences in the above samples are **0.76** and **0.04** for the baseline detector and our proposal, indicating that the impact of these identity words has been considerably reduced by our approach.

## 5 Conclusion

This work investigates biases in hate speech detection from lexical and sentence levels. Apart from the statistical correlation between artifacts and a specific label, we analyze the relative spuriousness of the feature based on its impact on local and global models. We find that most highly-correlated pattern features do not have high RS. After that, we analyze the generation process of HS biases from a causal view. We identify two confounders that cause the biases and propose Multi-Task Intervention from the model level and Data-Specific Intervention from the data level to mitigate them. Noticeable performance improvements on nine HS datasets and a label-balance challenge set indicate the effectiveness and robustness of our approach.

## Limitations

Our work is subject to a few limitations. First, our experiments are limited to English datasets. However, multilingual hate speeches are also frequently found on many social media platforms. There are still many challenges in languages other than English, especially some minor languages. A thorough examination of our methods' effec-

| Examples | Before | After |
|---|---|---|
| This is going to stir up a lot of yellow hatred for the hebrews . | hateful | non-hateful |
| I wonder if one of these guys is a Jew . | hateful | non-hateful |
| What you say about Blacks , Arabs and Jews are LIES. | hateful | non-hateful |
| Are they talking about the African women with huge lip plates? | hateful | non-hateful |

Table 4: Case study on white supremacist forum dataset. The identity words are highlighted. Before the intervention, the model makes mistakes on these non-hateful examples. Our proposed intervention can address such problems.

tiveness in languages other than English is necessary, which we leave as future work. Second, we use BERT-base-cased as the PLM backbone for most HS detectors following Ramponi and Tonelli, 2022, and we add GPT-3, ChatGPT and `text-davinci-003` as baselines. Other PLMs of different scales or architectures' robustness on various biases needs to be verified. Moreover, we only examine and mitigate biases in explicit HS datasets following previous works. However, biases in implicit HS (ElSherief et al., 2021b) are also important and we leave it as future work. In addition, our proposed RS and mitigation methods are applicable to any other text classification tasks, which we leave as future work. Besides, diverse prompt designs for counterfactual generators also need to be validated in future works. In this work, we only focus on single-turn HS detection tasks following previous works without considering user contexts (Yu et al., 2022), which is another practical setting. Finally, we find that some data is falsely annotated or based on different datasets. We do not modify them for fair comparisons. However, it is necessary to uniform the criteria for HS and correct problematic annotations. For token-level biases, although a large number of them are identity-related, there are still other highly-correlated tokens. We do not investigate them and leave it as future work.

## Ethics Statement

Our proposals in the research aim to mitigate biases and accurate detection of HS. The main datasets utilized in the study are open-access and publicly available. The offensive terms and identity-related words included as examples are mainly used to help researchers analyze the models more effectively. We do not involve annotators in this process given the sensitive nature of this work, and to reduce exposure of hateful content to human participants. We also add a content warning in the beginning of this paper to warn readers.

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

# A  Appendix

## A.1  Point-wise Mutual Information (PMI)

According to Gururangan et al., PMI is defined as following:

$$PMI(t, c) = \log \frac{p(t, c)}{p(t, \cdot)p(\cdot, c)}$$

We use this equation to find general (identity) biases in hate speech datasets.

## A.2  Dataset Statistics

Table 5 illustrates the statistics of datasets for MLM, MTL, and baseline models. The number of generated counterfactual examples after majority filtering are { MM: 631, WSF: 779, Fox: 558, HASOC: 872, ETHOS: 873, TSA: 250, AHS: 579, AOH: 1072, ZH: 520 }. Because many hate speech datasets do not have a specific split, we employ a 60-20-20 split for them.

## A.3  Details on OOD Datasets

We find that the phenomenon of label imbalance exists for many current HS datasets. Specifically, a large proportion of HS datasets contain much more hateful data instances than non-hateful ones in both train and test sets. As a result, to further mitigate the influence of such label imbalance, we construct a label-balance OOD challenge set containing five frequently occurring vulnerable groups, including Muslims, Women, LGBT, POC, and Jews.

| Dataset | # train | # dev | # test |
|---|---|---|---|
| MM | 445 | 149 | 149 |
| WSF | 6421 | 2141 | 2141 |
| Fox | 916 | 306 | 306 |
| HASOC | 3150 | 1171 | 1171 |
| ETHOS | 598 | 200 | 200 |
| TSA | 19176 | 6393 | 6393 |
| AHS | 14869 | 4957 | 4957 |
| AOH | 1932 | 645 | 644 |
| ZH | 14869 | 4957 | 4957 |

Table 5: Statistics of datasets in this work. These datasets are used in MLM, MTL and the main classification experiments.

Each group contains 15 hateful and non-hateful sentences. The total size of the challenge set is 150.

## A.4  Details for Counterfactual Generator

For token-level general biases (identity), we use the following example prompt for GPT-3.

> Generate 20 non-hateful sentences containing 'Jews':

For data-specific biases, we use the following prompt.

> Paraphrase the original sentence without using 'Trump is':
> Original sentence: Trump is the face of corporate greed. And the poor love him? Ironic.
> Paraphrased sentence:

We set the temperature to 1, max_tokens to 4000, top_p to 1.0, frequency_penalty to 1, and presence_penalty to 1 for GPT-3. The RS threshold we select is 0.2 for data-specific artifacts.

## A.5  Implementation Details

Following ElSherief et al., 2021b and Ramponi and Tonelli, 2022, we use a grid search to find the most suitable hyperparameters including learning rate [2e-5, 3e-5, 5e-5, 8e-6, 5e-6], batch size [4, 8, 10] and the number of epochs [1, 2, 3, 4]. We repeat every experiment three times using different random seeds and select the best models based on average macro $F_1$ on validation sets. Other hyperparameters are set to the default one in HuggingFace transformers' trainer. All experiments are

conducted on NVIDIA RTX 2080 Ti with 11GB memory. Every run of experiments can be finished within an hour.

## A.6   Set up for Baselines

### A.6.1   Prompt Design for GPT-3 Baselines

Following Chiu et al., 2021, we use the following prompts for LLMsbaselines:

**Zero-shot**:

> Is the following text hateful? Answer yes or no.
> 'Men can see beauty, so they don't want to sleep with black women or make goblin babies.'
> Answer:

**Few-shot**:

> Classify the following texts into 'non-hateful' or 'hateful':
> 'My best friend is muslim. And he is the best man in the world': non-hateful
> 'He seems like the very talkative taxi driver': non-hateful
> 'Men can see beauty, so they don't want to sleep with black women or make goblin babies.': hateful.
> 'Islam is a religion of hate. These propaganda type videos are dishonest. Simply read the Quran and Hadith.': hateful.
> 'Kill all muslims':

### A.6.2   Other Detail for Baselines

For AEDA, we generate four additional examples for augmentation for one original sentence. For LLMs' baselines, we randomly select 60 examples from the test sets and use the powerful text-davinci-002, text-davinci-003 and text-chat-davinci-002 through openai API and set the temperature to 0, max_tokens to 5, top_p to 1.0, frequency_penalty to 0, temperature to 0.7 and presence_penalty to 0. For MLM, we randomly mask 15% tokens and apply the default hyperparameters from the HuggingFace transformers trainer. Back translation is one of the commonly-used sentence-level DA methods. We use German as the intermediate language and Fairseq (Gehring et al., 2017) machine translation model pretrained on WMT19 (Foundation). We also try to combine two kinds of MTI including MTL and MLM by some weights. How-

ever, this combination can not yield a noticeable performance improvement.

## A.7   Examples of Data-Specific Biases

Apart from the categories in Section 4.5.1, a large number of data-specific biases can hardly be categorized.As a result, we do not list it on the tables. We listed Top-5 data-specific biases in Table 6.

| Root | Examples | MI | #Non-hate | #Hate | %Majority |
|------|----------|-----|-----------|-------|-----------|
| ② | ban this world cup, a and, better use to help, Trump is, wall that could have been | 0.16 | 19 | 557 | 96.42 |
| ③ | a.23 million waste, are a nation, a piece, at the bottom, used to celebrate a man, ... | 0.03 | 0 | 253 | 100.0 |
| ㉛ | ICC shift this world to another, called as national, pity on your, live in this, ... | 0.08 | 2 | 222 | 99.11 |
| ㉘ | the worst, a fucking, in the, Shift this, has a, ban Rain, ... | 0.03 | 35 | 167 | 82.67 |
| ⑮ | live in, him for, known as, sick of , tired of, called as, along with, ... | 0.03 | 50 | 188 | 78.99 |

Table 6: Examples of data-specific patterns in HASOC19. Root: the non-terminal root of PCFG. Examples: example grammar patterns of a certain root. MI: mutual information between the root and hateful label. #Non-hate: number of non-hateful sentences containing such grammar pattern. #Hate: number of hateful sentences containing such grammar pattern. %Maj: percentage of hateful examples.

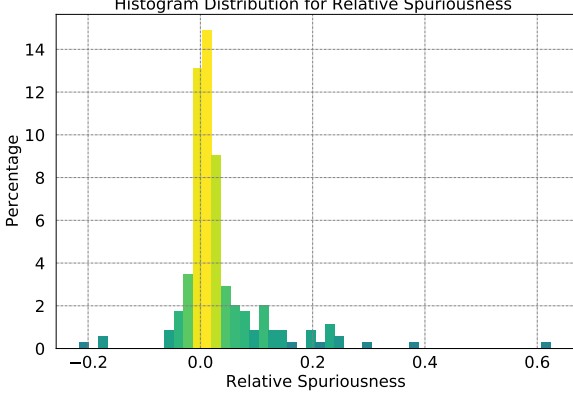

Figure 4: RS distribution of data-specific biases. A large proportion of data-specific artifacts remain positive, while most are around zero. This result indicates that most data-specific patterns like those in Appendix A.7 have a subtle influence on model predictions.