# OpenReview forum: "Mitigating Biases in Hate Speech Detection from A Causal Perspective"
_EMNLP/2023/Conference — EMNLP 2023 Findings_

### Official Review · Reviewer_CfXf · 2023-08-04

**Soundness:** 3

**Excitement:**

4: Strong: This paper deepens the understanding of some phenomenon or lowers the barriers to an existing research direction.

**Paper Topic And Main Contributions:**

This paper works toward the problem of mitigating biases in hate speech detection.
The authors propose a way to automatically identify and validate spuriousness.
They develop two methods to mitigate the biases by solving this problem from a causal perspective.
The results show consistent improvements in 9 hate speech datasets.


**Questions For The Authors:**

A. For spuriousness validation, why an unbiased model is equal to the average models trained on every HS dataset? Will there be situations where biases exist in every HS dataset? Why don't you use datasets from other domains for this step?

B. In line 388, what's the rationale that the authors choose to use majority vote?



**Reasons To Accept:**

1. The paper is overall well-structured.
2. The problem of hate bias does exist in current content moderation algorithms, and this paper works toward this problem.
3. As all the methods are automated, it is easier for future work to replicate and build on this work. The hate mitigation methods achieve good performance.
4. Studying this problem from a causal perspective is well-grounded and reasonable.

**Reasons To Reject:**

The main weakness of this paper is the metric choices and to me they are not well-justified in the paper.
It is unclear how an unbiased model can be considered as the average responses from models trained on every HS dataset.
The intuitions of the metric design may be better grounded on a theory or any previous work (line 254-268).

**Reproducibility:**

4: Could mostly reproduce the results, but there may be some variation because of sample variance or minor variations in their interpretation of the protocol or method.

**Reviewer Confidence:**

4: Quite sure. I tried to check the important points carefully. It's unlikely, though conceivable, that I missed something that should affect my ratings.

**Typos Grammar Style And Presentation Improvements:**

line 175: claim -> claims

Figure 3: I cannot see the black lines.

Line 386-389: I struggle to follow it and would suggest to add more explanations.

---

> ### Author Rebuttal · Authors · 2023-08-29
>
> ThThank you for your thoughtful review! We are grateful that you found our work’s research problem **important** and the **potential impact** of our method **on future work**. We are also pleased that you found out our method is **well-grounded from a causal perspective** and its **effectiveness with great performance**.
>
> - **W1: Metric choices are not well-justified**
>     - As described in **lines 404-409**, following **previous works**[1], we use macro-F1 as the main metric for evaluations, which is a **widely accepted metric** to evaluate the performance of an HS detection system. We acknowledge that it is not designed to directly evaluate how well biases are mitigated. However, as macro-F1 score gives a fair representation of the classifier's performance across **all** classes, which can reflect how biases are mitigated (prediction from hateful to non-hateful for non-hateful data containing short-cuts) to a certain degree. Besides, to make up for this limitation, we conduct a comprehensive **case study (Section 4.6 from line 541)** to analyze how well the biases are mitigated through our approaches both **qualitatively (Table 4)** and **quantitatively (line 565)**.
> - **Q1: Why do we treat the averaged model as an “unbiased model”? Will there be biases in every HS dataset? Why don't we use other domains’ data for this?**
>     - We treat the averaged model as an “unbiased model” under the assumption that different datasets are more likely to have different data-specific biases (adopted by **many other works** such as [2] to conduct cross-dataset analysis on spurious correlations).
>     - Yes, there will be biases in almost every HS dataset. As described in **Section 3.1.1 (line 206)**, the bias of identity words exists in almost all hate speech datasets. However, we use domain knowledge to find such biases following previous works [1][3][4] instead of Relative Spuriousness (RS). Besides, as shown in **Figure 3**, we only use RS to detect **data-specific biases (yellow part of the figure)**.
>     - Using datasets from other domains will deteriorate the BERT’s representation of hate speech. As a result, the classifier can not model the probabilities described in the expression from lines 271-275 well.
> - **Q2: What is the rationale for using a majority vote in line 388?**
>     - As described in **lines 385-387**, we use a majority vote to make sure the generated counterfactual data are non-hateful, which is a way to further **control the quilty** of the generated data.
> - Typos, Grammar, Style, and Presentation Improvements
>     - We will fix the typo in the caption of Figure 3 in the camera-ready version. The blue part indicates the framework for token-level biases (identity). The yellow part represents the framework for data-specific biases.
>     - We will add more detailed explanations such as the response to Q2 to Line 386-389 in the camera-ready version.
>
>
> [1] Alan Ramponi and Sara Tonelli. 2022. Features or Spurious Artifacts? Data-centric Baselines for Fair and Robust Hate Speech Detection. In Proceedings of the 2022 Conference of the North American Chapter of the Association for Computational Linguistics: Human Language Technologies, pages 3027–3040, Seattle, United States. Association for Computational Linguistics.
>
> [2] Tianlu Wang, Rohit Sridhar, Diyi Yang, and Xuezhi Wang. 2022. Identifying and Mitigating Spurious Correlations for Improving Robustness in NLP Models. In Findings of the Association for Computational Linguistics: NAACL 2022, pages 1719–1729, Seattle, United States. Association for Computational Linguistics.
>
> [3] Maarten Sap, Dallas Card, Saadia Gabriel, Yejin Choi, and Noah A. Smith. 2019. The Risk of Racial Bias in Hate Speech Detection. In Proceedings of the 57th Annual Meeting of the Association for Computational Linguistics, pages 1668–1678, Florence, Italy. Association for Computational Linguistics.
>
> [4] Maximilian Wich, Jan Bauer, and Georg Groh. 2020. Impact of Politically Biased Data on Hate Speech Classification. In Proceedings of the Fourth Workshop on Online Abuse and Harms, pages 54–64, Online. Association for Computational Linguistics.

---

### Official Review · Reviewer_W9Qi · 2023-08-05

**Soundness:** 3

**Excitement:**

4: Strong: This paper deepens the understanding of some phenomenon or lowers the barriers to an existing research direction.

**Missing References:**

cite the final published version of papers if possible e.g. 'Finding Dataset Shortcuts with Grammar Induction'

**Paper Topic And Main Contributions:**

- aims to reduce spurious correlation in hate speech detection.
- identifies token and sentence- level ones with pmi and pcfg respectively.
- reasoned with two types of confounders, which justified two types of intervention proposed: multi-task and data-specific(counterfactual learning).

**Questions For The Authors:**

what is the justification of using only the text of multimodal data? would the classification results still be meaningful?

**Reasons To Accept:**

- causal model justifies both proposed methods.
- each method was tested and proven effective against a variety of reasonable baselines.
- demonstrated good generalisation with new testing data.


**Reasons To Reject:**

- definition of hate speech is not clarified. debatable understanding of what hate speech is occasionally, mixing identity words and slurs, e.g. table 4 and section 4.5.2 , 'blacks' used as nouns to refer to people are not simply identity words and the sentences are hence not neutral. it is also questionable whether their influence to model prediction should be mitigated.
unclear, inconsistent definitions are already a big problem in this field. a paper with good methodologies but lack proper understanding of the task can be worrying.
- no information on quality of generated counterfactual data, e.g. any manual inspection, small batch annotation.

**Reproducibility:**

5: Could easily reproduce the results.

**Reviewer Confidence:**

4: Quite sure. I tried to check the important points carefully. It's unlikely, though conceivable, that I missed something that should affect my ratings.

**Typos Grammar Style And Presentation Improvements:**

- i think more people would agree that figure 1 top right trump example shouldn't be hateful than otherwise. either way when presenting examples, it's better to include source e.g. which dataset.
- expressions 272-275 are confusing, although the paragraph after it makes sense.  i suppose Pl(Y (X ̸= xi) = y|Xi = xi, Y = y) would be probability without feature xi but how does each element correspond to elements of that sentence? e.g. X ̸= xi would mean the input does not equal xi rather than not contain
- typos? vulnerable objectives -> identities
- many mentions of '9 datasets' before they are introduced. would be helpful to point to the dataset section when they are first mentioned

---

> ### Author Rebuttal · Authors · 2023-08-29
>
> Thanks for your thoughtful feedback! We appreciate that you find our proposed methods’ **great justification by causal models** and solid experiments with **sufficient baselines** and **in-depth generalization analysis**.
>
> - **W1: The definition of hate speech is not clarified.**
>     - We acknowledge that a clear definition of hate speech is important. However, it is a **challenging** task. In fact, the definition of hate speech is neither universally accepted nor are individual facets of the definition fully agreed upon[1]. As a result, different datasets may have subtly different criteria for hate speech. Our work uses **widely-used** public benchmarks following previous works [2][3][4].
> - **W1: No information on the quality of generated counterfactual data.**
>     - As described in **lines 385-391** and the **rightmost** part of **Figure 3**, to ensure the generated counterfactual data are non-hateful, we add majority-vote filtering after raw data generation. To further evaluate data quality, we conducted an **additional human study** on the generated counterfactual data (we randomly sampled 60 counterfactual data points from every dataset). Following Hartvigsen et al.[5], we first ask the annotators to guess whether the statement’s author was a human or an AI system. Then we ask annotators to measure the harmfulness of text on a **1-5 scale** with 1 being clearly benign and 5 indicating very hateful. The result of the human study is that **92%** of the generated data are considered human-generated by annotators. Besides, the average toxicity measure of **1.02** indicates that most of our generated data are non-hateful.
> - **Q1: What is the justification for using only the text of multimodal data?**
>     - We follow the practice of the **original Multimodal Meme Dataset paper**[6] where they conduct separate analyses on the textual part of their dataset using text-only models. As described in [6], the motivation for their practice is that the textual parts of this dataset are the captions of the memes, which contain **all the textual information** on the images which can be sufficient to determine if the meme is hateful for a large number of cases. (Their experiment results show **comparable** performances of text-only and multimodal classifiers.)
> - Typos, Grammar, Style, and Presentation Improvements
>     - The trump example in Figure 1 is from HASOC19[7] and is labeled as hateful. We will add the source in the Figure caption in the camera-ready version.
>     - The expression of $P_l (Y(X \neq x_i)| X_i = x_i, Y = y)$ is the probability of $y$ if $x_i$ is deleted for those that originally have the feature $x_i$ with the label $y$.
>
>
> [1] MacAvaney, Sean, et al. "Hate speech detection: Challenges and solutions." PloS one 14.8 (2019): e0221152.
>
> [2] Ilia Markov, Nikola Ljubešić, Darja Fišer, and Walter Daelemans. 2021. Exploring Stylometric and Emotion-Based Features for Multilingual Cross-Domain Hate Speech Detection. In Proceedings of the Eleventh Workshop on Computational Approaches to Subjectivity, Sentiment and Social Media Analysis, pages 149–159, Online. Association for Computational Linguistics.
>
> [3] Markov, T., Zhang, C., Agarwal, S., Eloundou Nekoul, F., Lee, T., Adler, S., Jiang, A., & Weng, L. (2023). A Holistic Approach to Undesired Content Detection in the Real World. Proceedings of the AAAI Conference on Artificial Intelligence, 37(12), 15009-15018.
>
> [4] Steve Durairaj Swamy, Anupam Jamatia, and Björn Gambäck. 2019. Studying Generalisability across Abusive Language Detection Datasets. In Proceedings of the 23rd Conference on Computational Natural Language Learning (CoNLL), pages 940–950, Hong Kong, China. Association for Computational Linguistics.
>
> [5] Thomas Hartvigsen, Saadia Gabriel, Hamid Palangi, Maarten Sap, Dipankar Ray, and Ece Kamar. 2022. ToxiGen: A Large-Scale Machine-Generated Dataset for Adversarial and Implicit Hate Speech Detection. In Proceedings of the 60th Annual Meeting of the Association for Computational Linguistics (Volume 1: Long Papers), pages 3309–3326, Dublin, Ireland. Association for Computational Linguistics.
>
> [6] Shardul Suryawanshi, Bharathi Raja Chakravarthi, Mihael Arcan, and Paul Buitelaar. 2020. Multimodal Meme Dataset (MultiOFF) for Identifying Offensive Content in Image and Text. In Proceedings of the Second Workshop on Trolling, Aggression and Cyberbullying, pages 32–41, Marseille, France. European Language Resources Association (ELRA).
>
> [7] Mandl, Thomas, et al. "Overview of the hasoc track at fire 2019: Hate speech and offensive content identification in indo-european languages." Proceedings of the 11th annual meeting of the Forum for Information Retrieval Evaluation. 2019.

---

### Official Review · Reviewer_5jqy · 2023-08-12

**Soundness:** 3

**Excitement:**

4: Strong: This paper deepens the understanding of some phenomenon or lowers the barriers to an existing research direction.

**Paper Topic And Main Contributions:**

The authors introduce the concept that artifacts related to specific labels are not confined solely to individual tokens; they can also extend to sentence-level elements, such as phrases within datasets. To explore this, they utilize a probabilistic context-free grammar tree to identify potential grammar candidates and employ pointwise mutual information (PMI) to highlight the most correlated grammar patterns. These patterns are identified as sentence-level (data-specific) spurious correlations. To validate their spuriousness and distinguish clean spurious correlations, they introduce a novel metric named "Relative Spuriousness." This is accomplished by measuring the disparity between local and global model scores, where the score is calculated as the difference between model results for inputs with and without the candidate spurious patterns. Additionally, they introduce "identity spurious," referring to identity words (tokens) with the highest PMI. Addressing two distinct types of bias, they present two mitigation methods. To counter identity bias at the token level, they again pre-train BERT through a masked language model (MLM) objective on nine hate speech datasets. They believe these token-level biases stem from biases in the underlying training data of pre-trained language models (PLMs). Subsequently, the retrained PLMs are fine-tuned on various hate speech tasks simultaneously, resulting in superior performance compared to baseline methods for different hate speech datasets. The second method (DSI) involves counterfactual learning, generating non-hate speech samples containing identified biases (both token-level and sentence-level) using GPT. By fine-tuning the model on this augmented dataset, they achieve improved performance in comparison to other augmentation techniques.






**Questions For The Authors:**

Question A: I'm confused about the MTI method. Does it refer to applying both MLM and then MTL sequentially? Or is each of these two independently referred to as MTI? If the first case is correct, why wasn't the result of MTI (MLM + MTL) reported in table 1?

**Reasons To Accept:**

The proposed methods effectively enhance the performance of hate speech (HS) tasks  mitigating the biases and exhibit creativity in their approach. The authors utilize GPT to generate non-hate speech data containing the identified biases, which serves as a means to mitigate biases. Overall, both proposed methods are valuable and impactful, leading to improvements. Through some straightforward steps, they manage to elevate the model's performance in tackling the challenging task of hate speech detection. They also introduce a novel method to identify sentence-level spurious.

**Reasons To Reject:**

Considering that they introduce a new type of potential bias (sentence-level) and propose a metric to measure their spuriousness, it would be essential  to validate and show the accuracy of these metrics. Since the RS distribution figure suggests that these biases exert a subtle influence, it's possible that the improvement achieved by the DSI method is mainly due to the mitigation of token-level biases . I believe that the DSI experiment should be analyzed separately for both sentence-level and token-level counterfactual generated examples to see the effectiveness of mitigating biases with high RS.

**Reproducibility:**

4: Could mostly reproduce the results, but there may be some variation because of sample variance or minor variations in their interpretation of the protocol or method.

**Reviewer Confidence:**

3: Pretty sure, but there's a chance I missed something. Although I have a good feel for this area in general, I did not carefully check the paper's details, e.g., the math, experimental design, or novelty.

**Typos Grammar Style And Presentation Improvements:**

I would appreciate the inclusion of statistics regarding the correlation between detected token or grammar biases and the corresponding labels in the mentioned datasets. It would be beneficial to have this information, even if it's presented in an appendix.

In lines 32 and 33, the reference (Re033 lia et al., 2019) is inaccurately addressed, as it pertains to the arXiv version instead of the published version in AAAI. This issue also occurs in line 240. While I haven't checked all the references, it's advisable to verify that all references are cited with their original published versions to ensure accuracy.

In lines 491 and 492, you mentioned using MTL as a baseline in Table 3, but in Table 3, you wrote MTI. This is confusing.

The caption of Figure 3 mentions the presence of black and blue lines, but there are no such lines visible in the actual figure.


Line 102 mentions "as shown in Figure 3.2,"; however, Figure 3.2 is not a direct reference to the expected figure, which I assume might be Figure 2. Instead, it refers to the section that discusses the content related to it.

I believe it would be beneficial to provide a more comprehensive explanation behind the choices of G1 and G2, along with the rationale for applying the proposed methods to mitigate their effects.

---

> ### Author Rebuttal · Authors · 2023-08-29
>
> Thanks for your insightful and positive feedback! We are encouraged that you found our proposed mitigation methods **valuable** and **impactful** for challenging hate speech detection. We are also glad that you highlighted the **novelty** of our proposed metric of “Relative Spuriousness” to identify sentence-level spuriousness which was ignored by previous works and the **detailed statistics** on the correlation between detected token or grammar biases.
>
> - **W1: The DSI experiment should be analyzed separately to validate the effectiveness of biases high RS (sentence-level biases).**
>     - We appreciate your constructive feedback and acknowledge the importance of validating DSI’s effectiveness independently. In response to your comments, we have conducted the following **additional experiments** to **separately analyze** the DSI’s performance. To be specific, for sentence-level biases, we collect the non-hateful data points that contain highly biased grammar patterns (see Table 6 as examples) towards the label of hateful in the test set as the evaluation of the sentence-level bias mitigation. Similarly, for token-level biases, we collect the non-hateful data points that have highly correlated identity words to evaluate token-level bias mitigation. We then train BERT with the original training set and with DSI augmentation respectively. We report the average difference in the accuracy of these two methods in the following table:
>
>         | Dataset | MM | WSF | Fox | HASOC | ETHOS | TSA | AHS | AOH | ZH | Average |
>         | --- | --- | --- | --- | --- | --- | --- | --- | --- | --- | --- |
>         | Acc difference (sentence-level) | 20.93 | 8.54 | 11.43 | 23.52 | 17.39 | 7.69 | 14.28 | 12.97 | 5.08 | 13.54 |
>         | Acc difference (token-level) | 25.58 | 38.89 | 21.42 | 13.79 | 35.89 | 17.39 | 23.21 | 12.50 | 14.46 | 22.57 |
>
>         The experiment result indicates that DSI can **noticeably mitigate** sentence-level token-level biases. Comparatively, token-level DSI is more effective.
>
> - **Q1: Does MTI (Multi-Task Intervention) refer to MLM (Masked Language modeling) and then MTL (Multi-Task Learning) sequentially or those two independently?**
>     - As described in the paraphrase starting from **line 344**, MLM and MTL are two types of MTI, which are independently presented in our experiments. Precisely, MTI is an abstract Intervention derived from causal analysis while MLM and MTL are two solutions to achieve such Intervention. We will further clarify these concepts in the following camera-ready version.
> - Typos, Grammar, Style, and Presentation Improvements
>     - For the MLI in Table 3, it is a typo and we will fix it to MTL as described in lines 491 and 492. We will fix all the citations from the arXiv version to the published version and other typos (Figure 3’s caption and line 102) in the camera-ready version.
>     - **The rationale behind the choices of G1 and G2 with the proposed method to mitigate them.**
>         - The choice of G1 comes from the **pre-training data** of PLMs. As the pre-training data of PLMs such as BERT are mostly from books or the internet, the distribution of different identities can be hugely diverse. For example, the identity words of minority groups can occur several orders of magnitude less frequently than other groups. As a result, such imbalance distribution (G1) can bring biases to PLMs. To mitigate such biases from the pretraining data, we propose MTI of training auxiliary tasks using a diverse range of HS datasets under the assumption that minorities occur more frequently in HS datasets.
>         - The choice of G2 comes from **hate speech training data**. In most HS data, the distribution of different identities in hateful and non-hateful contexts is not even (see **lines 336-339** as an example, G2 can be the platform of data source). As a result, training with such data can bring biases to certain identities such as Muslims or Jews. To mitigate such biases from the HS training data, we propose DSI to augment the current HS dataset with counterfactual data from both token-level and sentence-level to change the original distribution.

---

### Meta-Review · Area_Chair_Q6ZZ · 2023-09-19

**Recommendation:** 3

**Metareview:**

This paper proposes an approach to mitigate bias in hate speech detection approaches by automatically finding likely non-causal correlates.

Overall strengths include the focus on a current and important problem, a well-motivated approach with a potentially novel "spuriousness" measure, and good performance across multiple datasets at least for the metrics shown. Weaknesses include: choice of evaluation metrics, lack of validation for non-causal confounds/sentence-level spuriousness, and a lack of clear definition of the task (hate speech detection).

---

### Decision · Program_Chairs · 2023-10-07

**Decision:**

Accept-Findings

**Comment:**

This paper proposes an approach to mitigate bias in hate speech detection approaches by automatically finding likely non-causal correlates.

Overall strengths include the focus on a current and important problem, a well-motivated approach with a potentially novel "spuriousness" measure, and good performance across multiple datasets at least for the metrics shown. Weaknesses include: choice of evaluation metrics, lack of validation for non-causal confounds/sentence-level spuriousness, and a lack of clear definition of the task (hate speech detection).